# Evaluation of Anomalies and Neurodevelopment in Children Exposed to ZIKV during Pregnancy

**DOI:** 10.3390/children9081216

**Published:** 2022-08-12

**Authors:** Kathia Guardado, Miguel Varela-Cardoso, Verónica Ofelia Pérez-Roa, Jaime Morales-Romero, Roberto Zenteno-Cuevas, Ángel Ramos-Ligonio, Oscar Guzmán-Martínez, Clara L. Sampieri, Christian S. Ortiz-Chacha, Rosybet Pérez-Varela, Cristina Fernanda Mora-Turrubiate, Hilda Montero

**Affiliations:** 1Instituto de Salud Pública, Universidad Veracruzana, Xalapa 91090, Mexico; 2Centro de Investigaciones Biomédicas, Universidad Veracruzana, Xalapa 91090, Mexico; 3Facultad de Medicina, Universidad Veracruzana, Ciudad Mendoza 94740, Mexico; 4Hospital Regional de Río Blanco, Secretaría de Salud de Veracruz, Río Blanco 94733, Mexico; 5LADISER Inmunología y Biología Molecular, Facultad de Ciencias Químicas, Universidad Veracruzana, Orizaba 94340, Mexico; 6Centro de Fisioterapia Nirho, Orizaba 94300, Mexico

**Keywords:** Zika, ZIKV, congenital Zika syndrome, major anomalies, minor anomalies, neurodevelopmental, EDI, DDSST-II

## Abstract

Zika virus (ZIKV) infection in pregnancy is associated with birth and developmental alterations in infants. In this study, clinical records of 47 infants whose mothers had Zika during pregnancy or clinical manifestations compatible with Zika were reviewed. A description of the infants’ anomalies was established, and a neurodevelopmental assessment was performed on 18 infants, using the Evaluation of Infant Development (EDI for its initialism in Spanish) and DDST-II (Denver Developmental Screening Test II) tests. From his sample, 74.5% of the infants evaluated had major anomalies and 51.9% had minor anomalies. The incidence of major anomalies, related to trimester of pregnancy, was 84.2% for the first trimester, 77.8% for the second trimester, and 37.5% in the third trimester. A similar trend was observed in the frequency of infants without anomalies and was less evident in the incidence of minor anomalies (*p* = 0.016). Through neurodevelopmental assessments, EDI identified 27.8% of infants as having normal development, while 55.5% of affected infants had developmental delay, and 16.7% were at risk for developmental delay. The DDSST-II showed that 77.7% infants had delay in the gross motor and language area, 88.8% in the fine-adaptative motor area, and 72.2% in the personal–social area. In this work, children of mothers with ZIKV infection during pregnancy may have major or minor anomalies regardless of the trimester of pregnancy in which the infection occurred. The neurodevelopmental assessment shows that ZIKV can cause a developmental delay in infants with the fine-adaptative motor area being the most affected.

## 1. Introduction

Zika virus (ZIKV) is a virus different from other arboviruses. One of these particularities is that mosquito bite is not its only route of transmission, it can be transmitted sexually [1], from mother to child during pregnancy [2], and through blood transfusion [3]. ZIKV infection can cause Zika disease, characterized by rash, pruritus, arthralgia, headache, myalgia, and fever, among others [4]. However, it is estimated that ZIKV infection may be asymptomatic in 80% of the infected population [5].

Outbreaks of Zika virus ZIKV in humans were sporadic until early 2015, when the first cases were confirmed in Brazil [6]. ZIKV spread rapidly in the Americas, with active virus transmission reported in 38 countries on the American continent [7]. The importance of ZIKV made it a worldwide public-health emergency decreed by the World Health Organization (WHO) in 2016 [8]. In Mexico, ZIKV was detected for the first time in 2015 and since then, and until the end of 2021, 55.1% of the reported cases had been in pregnant women [9]. It is estimated that 14% of children exposed to ZIKV during gestation may develop health problems associated with infection, including neurodevelopmental disorders [10]. The high incidence of cases in pregnant women is important due to the fact that ZIKV infection during pregnancy has been associated with the development of anomalies in newborns [11], such as morphological–cranial [12], cerebral [13,14,15], ophthalmic [16], neurological [17], and joint contractures [18,19], which have been grouped into a new syndrome called Congenital Zika Syndrome (CZS) [11,20]. Some anomalies caused by intrauterine ZIKV infection can manifest postnatally, such as postnatal microcephaly [21] and neurodevelopmental disorders, that could affect cognitive, motor, and social functions [22], even in healthy infants [23]. Neurodevelopmental assessment in infants exposed to ZIKV during pregnancy is of great importance because it allows the characterization of the effects of gestational ZIKV infection at birth and in the long term, due to the possible impairment of cognitive, motor, and social skills in infants. Characterization of neurodevelopmental disorders should be performed by standardized tests according to the age of the infant [24].

ZIKV infection has been important to Mexico since some studies have found cases of infants exposed to ZIKV during gestation with signs of CZS [25,26]. In normocephalic Mexican children, it has also been observed that exposure to ZIKV could affect neurodevelopment by decreasing the infant’s fine and gross motor skills, such as visual reception, and receptive language up to 6 months of age [27]. The aim of this study is to evaluate abnormalities and neurodevelopment of infants with a mean age of 33 months born to mothers infected with ZIKV during pregnancy. The sample was taken in a public institution in the state of Veracruz, Mexico, which provides care to low-income children with different disabilities.

## 2. Materials and Methods

### 2.1. Study Design

This study was approved by the Research Ethics Committee of the Institute of Public Health Research Center (CEI-ISP-R02/2020), the Ministry of Health (SEIC-004-18), and authorities of the Center for Infant Rehabilitation of Veracruz (CRIVER for its acronym in Spanish). A retrospective cohort study including 47 children whose mothers presented symptoms compatible with Zika during pregnancy, was analyzed. Participants were grouped into confirmed cases considering any probable case of ZIKV infection confirmed by laboratory techniques approved by health authorities, and probable cases considering any pregnant woman presenting two or more symptoms compatible with Zika and with a history of visiting or residing in areas endemic for the ZIKV. This was performed according to definitions used by the Mexican Ministry of Health (SSA) [28], similar to other studies [18,29].

### 2.2. Clinical Data Collection

The variables of interest were collected using an instrument developed by the research team. This instrument comprises anthropometric measures and descriptive variables of the mother and the infant that are part of the study.

The medical records of the infants were reviewed to determine the possible anomalies that occurred at birth, taking into consideration: (a) low weight: newborns weighing < 2500 g, (b) low size: newborns with growth patterns below gestational age according to the parameters of the Fenton growth chart provided by the Pan American Health Organization (PAHO), (c) prematurity: newborns born at <37 weeks pregnancy, and (d) decreased head circumference: determined according to gestational age following the parameters of the Fenton growth chart provided by PAHO [30]. The anomalies reported in the development of the infants were divided into six categories: cranial morphology, cerebral alterations, ophthalmological alterations, joint contractures, neurological sequelae, and others, similar to other studies [11,20]. The anomalies were classified into major and minor anomalies based on the Centers for Disease Control and Prevention (CDC) definitions of congenital anomalies, indicated as: major anomalies to those anomalies of body structure or function that account for the majority of deaths, morbidity, and disability related to congenital anomalies; and minor anomalies to those anomalies of structure or function that pose no significant health problems in the neonatal period and tend to have limited social or cosmetic consequences for the affected individual [31]. For the obstetric risk maternal age variable, risk ages were considered: adolescent pregnancy ≤ 19 years [32] and geriatric pregnancy ≥ 35 years [33].

### 2.3. Neurodevelopmental Assessments

Two neurodevelopmental assessments were applied to 18 infants: (1) EDI [34], developed by the National Institute of Perinatology of Mexico, to be used for using in Mexican infants under 5 years of age to detect neurodevelopmental problems, and (2) DDST-II, used to evaluate the development of infants from 0 to 6 years in the gross-motor, fine-adaptive motor, language, and personal–social areas [35]. The tests were applied by a pediatrician and a neuropediatrician.

### 2.4. Statistical Analysis

The categorical variables were described as absolute frequencies and percentages, while age was described as the mean and standard deviation. Comparison between categorical variables was performed using the chi-square test for trends or Fisher’s exact test. The probability of infants having developmental abnormalities was expressed as cumulative incidence. Statistical significance was defined as a value of *p* ≤ 0.05. Statistical analysis was performed with SPSS (IBM SPSS Statistics for Mac Version 21.0. Armonk, NY) and StatCalc (Epi Info; CDC, Atlanta, GA, USA).

## 3. Results

### 3.1. Characteristics of Pregnant Women and Effects of ZIKV Infection at Birth

Forty-seven infants whose mothers presented signs and symptoms compatible with ZIKV during pregnancy were included. The mothers were classified into two groups, confirmed cases (*n* = 20) and probable cases (*n* = 27), with a mean age of 24.8 ± (5.4) and 23.4 ± (6.4) years, respectively. The sociodemographic characteristics of the mothers are described in Table 1. According to the trimester of pregnancy for confirmed cases, ZIKV infection was detected in 50% (*n* = 10) in pregnant women in their first trimester, while 25% (*n* = 5) was found in pregnant women in their second trimester and 25% (*n* = 5) were in the third trimester. In probable cases, 7.4% (*n* = 2), it was not possible to determine the trimester in which the infection occurred. Of the documented cases, 33.3% (*n* = 9) were in the first trimester, 48.1% (*n* = 13) were in the second, and 11.1% (*n* = 3) were in the third (Table 1).

In the studied newborns, 20% (*n* = 4) of the confirmed cases presented low weight; 40% (*n* = 8) low size; 15% (*n* = 3) prematurity; and 4 % (*n* = 9) decreased head circumference. In probable cases, 18.5% (*n* = 5) had low weight; 51.9% (*n* = 14) low size; 14.8% (*n* = 4) prematurity; and 70.3% (*n* = 19) decreased head circumference (Table 1).

### 3.2. Presence of Major and Minor Anomalies in Infants

Information was collected on the anomalies reported in infants and classified into six categories: cranial morphology, cerebral alterations, ophthalmological alterations, joint contractures, neurological sequelae, and others, similar to other studies [20], to categorize them into major and minor anomalies [31].

The most frequent major anomalies in confirmed cases of ZIKV infection were spastic quadriplegia in 60% (*n* = 12); microcephaly in 50% (*n* = 10); and multiple joint contractures in 45% (*n* = 9). Similarly, the most frequent major anomalies in probable cases were microcephaly in 74.1% (*n* = 20); spastic quadriplegia in 66.7% (*n* = 18); and multiple joint contractures in 25.9% (*n* = 7) (Figure 1). The most frequent minor anomalies in the confirmed cases of ZIKV infection were delayed psychomotor development in 60% (*n* = 12); cognitive impairment in 25% (*n* = 5); and diffuse calcifications in 25% (*n* = 5). As for the probable cases, delayed psychomotor development in 88.9% (*n* = 24); cognitive impairment in 63% (*n* = 17); and seizures in 40.7% (*n* = 11) (Figure 1). Of the infants, 4.3% (*n* = 2) died and 5% (*n* = 1) of the infants of confirmed with ZIKV and 3.4% (*n* = 1) of probable cases developed postnatal microcephaly.

An inverse relationship was found between the cumulative incidence of major anomalies and the trimester of pregnancy (Table 2). Mothers in the first trimester of pregnancy presented the highest incidence (84.2%), followed by those in the second trimester with (77.8%), and finally, an incidence of (37.5%) for those in the third trimester (*p for trend* = 0.016). Although this relationship was not as evident when comparing the incidences of minor anomalies. A similar trend was observed when the frequency of infants without anomalies increased in relation to the trimester of pregnancy: first trimester (0%), second trimester (16.7%), and third trimester (25.0%). Concerning maternal age as a factor for the development of major or minor anomalies in infants, no significant difference was found in the incidence of major or minor anomalies between women with and without obstetric risk age (91.7% vs. 88.6%, respectively) (*p* = 0.99) (Table 2).

### 3.3. Neurodevelopmental Assessment

Of the infants included in this study, 53.2% abandoned their care process on CRIVER; therefore, the neurodevelopmental evaluation was performed in 18 infants, with a mean age of 33.1 ± (6.4) months. The EDI and DSST-II tests were applied; with the DDSST-II test, the percentage of infants showing delay was determined according to the four areas evaluated (gross-motor, fine-adaptive motor, language, and personal–social) to observe the delay presented in the population (Table 3). Moreover, 27.8% of the infants presented normal neurodevelopment and 72.2% presented some risk or delay in one of the developmental areas evaluated: 77.7% (*n* = 14) showed delay in the gross-motor area; 88.8% (*n* = 16) delay in the fine-adaptative motor area; 77.7% (*n* = 14) delay in language area; and 72.2% (*n* = 13) delay in the personal–social area (Table 3). Through the EDI test, it was determined that 27.8% (*n* = 5) showed normal development while 55.5% (*n* = 10) and 16.7% (*n* = 3) of the infants evaluated presented risk of developmental delay and developmental delay, respectively (Table 3).

## 4. Discussion

ZIKV is an arbovirus of recent circulation in Mexico [36]. In contrast to other arboviruses, ZIKV is teratogenic and its effects on infants were addressed in various research studies [37,38,39,40]. CZS is characterized by birth and developmental abnormalities that are currently not fully understood [40,41,42]. This study analyzed the characteristics in children of mothers who were infected with ZIKV during pregnancy, at birth, and during their development.

In our study population, 55% of the newborns of confirmed cases and 29.7% of probable cases were born healthy. Microcephaly was one of the most frequent alterations found at birth, as in other studies [37,41]. In addition, two cases of postnatal microcephaly were identified, a characteristic that has been previously described [21], which highlights the importance of following infants who were exposed to intrauterine ZIKV infection. The classification of major and minor anomalies in infants with intrauterine ZIKV infection allowed us to observe their possible negative impact during development. The most frequent major anomalies in infants of this study, in addition to microcephaly, were spastic quadriplegia and multiple joint contractures, previously described as part of the pattern of signs for CZS [11]. The most frequent minor anomalies in the infants in this study, both in confirmed and probable cases, were delayed psychomotor development and cognitive impairment, related to neurodevelopmental alterations. The description of motor impairments in infants with ZIKV exposure in utero has been reported [43]. The diagnosis of some of the minor anomalies, such as those observed in this study, needs to be characterized through neurodevelopmental assessment tests. Because of this, the inclusion of these tests should be considered important in the care of children exposed to ZIKV in pregnancy.

We also evaluated the incidence of anomalies according to the pregnancy trimester in which ZIKV infection occurred. As in other studies, we found that the highest incidence of major anomalies is found in the first trimester [44,45,46]. However, in this study, the incidence of developing major anomalies is latent throughout pregnancy. Regarding minor anomalies, the highest incidence was found in the third trimester group. It was also observed that the frequency of infants without anomalies increased as the pregnancy progressed, indicating that infants born to ZIKV-infected mothers in late pregnancy are less likely to develop anomalies. Maternal age was also analyzed, with the incidence of anomalies in newborns; however, no significant association was found, suggesting that the development of these anomalies could be intrinsic to gestational ZIKV infection, in agreement with what has been previously reported [41]. The development of major and minor anomalies is present throughout pregnancy, and although minor anomalies are not life-threatening for the infant during development [31], they may have an impact on the infant’s quality of life and family. Therefore, the early detection and characterization of these minor anomalies could allow the implementation of appropriate therapies and treatments to help improve the quality of life of the child. These data highlight the need for measures to prevent infection throughout pregnancy, regardless of the trimester in which the infection occurred.

On the other hand, SCZ is a new syndrome, so its characterization is important and for which neurodevelopmental assessments are a valuable tool. In previous reports of infants exposed to ZIKV during pregnancy, it was observed that the risk of developing neurodevelopmental disorders may be related to the presence of microcephaly [22,47]. The risk of neurodevelopmental delay can range from 13.8% to 20.2% in normocephalic infants, while this percentage increases from 99.1% to 100% when infants develop severe microcephaly or from 65% to 70% in cases of moderate microcephaly [48]. Our results show that 72.2% of the infants studied, with and without anomalies, may present developmental risk or delay. This percentage could be high considering that our population includes infants without microcephaly. More studies are needed to elucidate the role of structural and physical abnormalities in the functional impairment of the infant. The DDST-II test was used to evaluate infants exposed to intrauterine cytomegalovirus and rubella virus infections [49,50]. Intrauterine cytomegalovirus exposure caused cognitive/developmental impairment and motor delay in 16.4% and 7.3%, respectively [49]. The same test, in cases of infants exposed to rubella virus intrauterine, showed language delay in 69.2%, impairment of the gross-motor area in 46.2% of cases; of the fine-adaptive motor area in 30.8 %; and the personal–social area in 46.2% [50]. The infants in this study presented greater damage, since 88.8% of the infants presented impairment in the fine-adaptive motor area; 77.7% in gross motor and language delay; and 72.2% in the personal–social area. Our results suggest that ZIKV could be one of the most teratogenic viruses severely affecting neurodevelopment in infants.

Two neurodevelopmental assessments tests were used in our study: EDI and DDST-II. Although the tests evaluate different aspects, the results of both show correlation in the developmental damage of the infant. In Mexico, there is already a validated test for the national population that could be used in a similar way to the DDST-II and compared with the results of other international studies [51,52]. One of the limitations of our study is it only studied children of ZIKV-infected mothers who had some alteration and who attended a medical center for the care of children with disabilities which may make the frequency of anomalies higher than that reported in other studies [44,53]. Another limitation of our study was the small sample size, however, our findings suggest that the risk of developing anomalies in newborns with gestational Zika is present throughout pregnancy and propose the establishment of a management program for children of mothers infected with ZIKV during gestation similar to other teratogenic viruses, such as CMV [54] and rubella [55]. Further, prevention and control measures for ZIKV infection in pregnant women should be implemented throughout pregnancy.

## 5. Conclusions

Our study evaluates for the first time the abnormalities and neurodevelopment in infants, with and without microcephaly, exposed to ZIKV during pregnancy in Mexico at 33 months of age. This work highlights the importance of studying infants exposed to ZIKV during pregnancy to characterize CZS because, as a new syndrome, the clinical manifestations have not been fully described. Our study suggests that ZIKV could be teratogenic, regardless of the trimester of pregnancy in which the infant was exposed to the virus. However, future studies with a larger population are needed to corroborate these findings. If these findings are consistent with future studies, prenatal management guidelines should be modified to include ZIKV screening of pregnant women in ZIKV-endemic areas, as well as follow-up and neurodevelopmental evaluation of ZIKV-exposed infants.

## Figures and Tables

**Figure 1 children-09-01216-f001:**
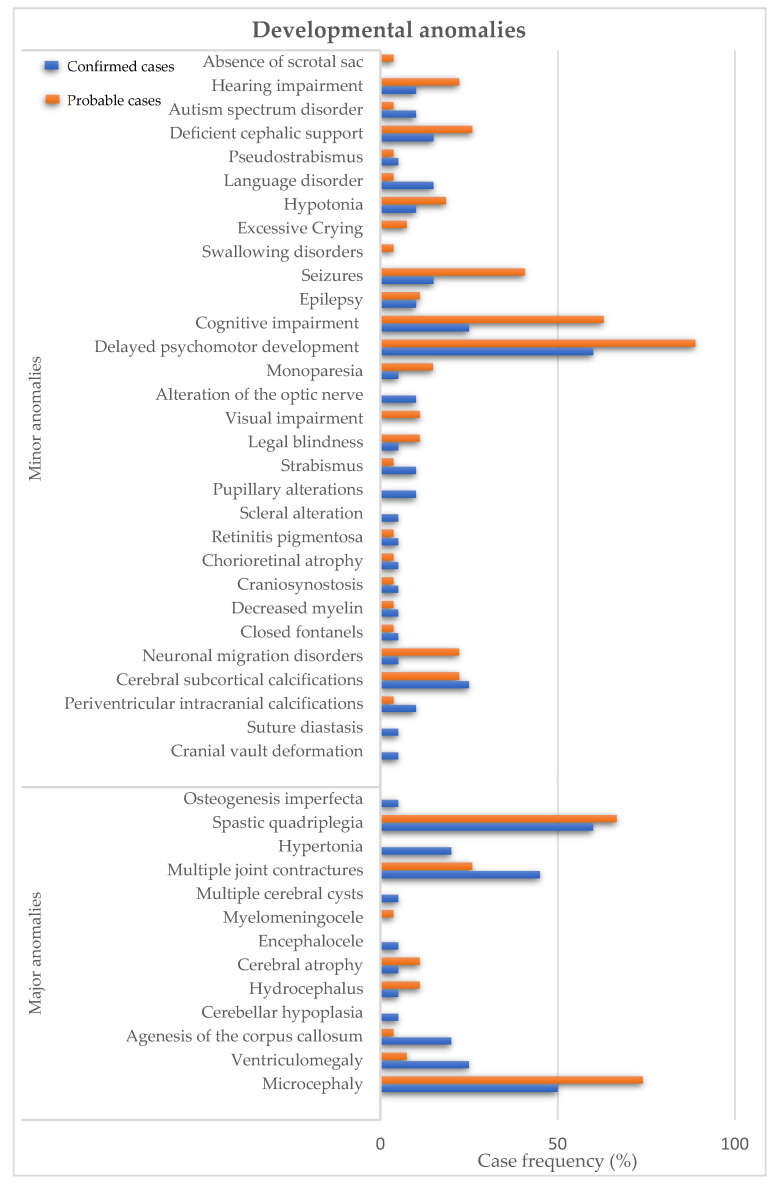
Developmental anomalies in children of mothers infected with ZIKV during pregnancy. Major anomalies are described in confirmed cases (blue bars) and probable cases (orange bars).

**Table 1 children-09-01216-t001:** General characteristics of mothers during pregnancy and of the infant at birth for confirmed and probable cases of ZIKV.

Characteristics	Confirmed Cases *n* = 20	Probable Cases *n* = 27
Age (years), mean ± SD	24.8 ± (5.4)	23.4 ± (6.4)
Education, *n* (%)		
Elementary school	0	2 (7.4)
Middle school	6 (30)	7 (25.9)
High school	4 (20)	11 (40.7)
University	8 (40)	6 (22.2)
No information	2 (10)	1 (3.7)
Residence, *n* (%)		
Veracruz rural area	2 (10)	3 (11.1)
Veracruz urban area	17 (85)	23 (85.2)
No information	1 (5)	1 (3.7)
Blood type, *n* (%)		
O+	15 (75)	10 (62.5)
A+	2 (10)	5 (31.3)
B+	0	1 (6.2)
No information	3 (15)	11 (40.7)
Clinical and obstetric variables, *n* (%)		
Miscarriage	1 (5)	4 (14.8)
TORCH+	3 (15)	5 (18.5)
Cervical-urinary tract infections	3 (15)	2 (7.4)
Toxicomania	3 (15)	0
Type of delivery, *n* (%)		
Cesarean delivery	14 (70)	20 (74.1)
Natural childbirth	5 (25)	7 (25.9)
No information	1 (5)	0
Trimester of ZIKV infection, *n* (%)		
First trimester	10 (50)	9 (33.3)
Second trimester	5 (25)	13 (48.1)
Third trimester	5 (25)	3 (11.1)
No information	0	2 (7.4)
Clinical characteristics of the newborn at birth, *n* (%)		
Low weight	4 (20)	5 (18.5)
Low size	8 (40)	14 (51.9)
Prematurity	3 (15)	4 (14.8)
Decreased head circumference	9 (45)	19 (70.3)

SD: Standard deviation. ZIKV: Zika virus.

**Table 2 children-09-01216-t002:** Cumulative incidence of anomalies according to trimester of pregnancy and maternal age at the time of ZIKV infection.

	With Major Anomalies *n* (%)	With Minor Abnormalities Only *n* (%)	No Anomalies *n* (%)	*p*-Value
Trimester of pregnancy				
First trimester, *n* = 19	16 (84.2)	3 (15.8)	0 (0.0)	0.016 *
Second trimester, *n* = 18	14 (77.8)	1 (5.5)	3 (16.7)	
Third trimester, *n* = 8	3 (37.5)	3 (37.5)	2 (25.0)	
Maternal age with obstetric risk ^§^				
Yes, *n* = 12	9 (75.0)	2 (16.7)	1 (8.3)	0.99 **
No, *n* = 35	26 (74.3)	5 (14.3)	4 (11.4)	

ZIKV: Zika virus. Some infants with major anomalies also had minor anomalies. ^§^ Maternal age with obstetric risk was defined as <19 and >35 years. Reference group without risk: 20 to 34 years. * The chi-square test for trend was used. ** Fisher’s exact test was used in a 2 × 2 table by regrouping in a single group those with major anomalies or minor anomalies according to maternal age with risk (*n* = 11) or without obstetric risk (*n* = 31).

**Table 3 children-09-01216-t003:** Neurodevelopmental assessment in children exposed to ZIKV infection during pregnancy.

Participant	DDST-II						EDI	
	Chronological Age	Gross Motor	Fine-Adaptive Motor	Language	Personal–Social	Psychomotor Development Score	Neurological Age Equivalent	Test Outcome
Participant 1	34	26	24	14	16	1	25 to 30	
Participant 2	36	7	7	7	5	2	7 to 9	
Participant 3	36	36	34	36	36	0	36	
Participant 4	36	36	36	36	36	0	31 to 36	
Participant 5	36	36	35	38	36	0	36	
Participant 6	35	36	36	36	36	0	31 to 36	
Participant 7	34	13	14	10	16	2	10 to 12	
Participant 8	32	4	3	4	4	2	4	
Participant 9	35	24	24	18	24	2	26	
Participant 10	40	34	34	34	34	1	36	
Participant 11	33	30	30	30	33	0	31	
Participant 12	36	7	5	8	8	2	5 to 6	
Participant 13	34	4	4	5	5	2	4	
Participant 14	34	2	2	2	2	2	1	
Participant 15	34	28	24	28	28	1	30	
Participant 16	14	7	9	7	9	2	7 to 9	
Participant 17	32	6	6	7	6	2	3	
Participant 18	35	8	8	9	9	2	10 to 12	

Ages are represented in months. ZIKV: Zika virus. DDST-II: Denver Developmental Screen Test II. EDI: Evaluation of Infant Development. Psychomotor development score (DDST-II): 2: Delay, 1: Risk, 0: Normal, Test outcome (EDI): 

 Risk of developmental delay: the child does not adequately achieve the developmental milestones and skills expected for his/her age group and is significantly delayed because he/she has not achieved the milestones of the previous age group or has high-risk signs such as red flags or has an abnormal neurological examination. 
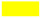
 Lagging in development: the child does not achieve all the milestones and skills expected for his/her age group but is not significantly delayed because he/she has achieved the milestones of the previous age group. 
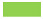
 Normal development: the infant achieves the milestones and skills expected for his/her age group in all areas of development without presenting alarm signals or abnormal data in the neurological examination.

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
