# Peer review of "Evaluation of Anomalies and Neurodevelopment in Children Exposed to ZIKV during Pregnancy"

_children, 2022, doi:10.3390/children9081216_

Round 1

Reviewer 1 Report

Dear Authors

Very important work on the neurodevelopmental disorders in children exposed intrauterine to zika virus. However, you must pay attention to the following points:

Very short introduction section. You made a sharp introduction to the background of the study. Better start by mentioning the pandemic, the areas where the pandemic was, the way of transmission and then with the main symptoms. The prevalence in affected areas…

As important as the findings are, the sample is too small to invalidate previous research results. It is therefore important in a paragraph after the discussion to highlight the limitations and strengths of the study.

The conclusions should emphasize the importance of conducting similar studies with a larger population in order to reach safe conclusions.

Author Response

Dear Reviewer:

We thank you for reviewing our manuscript, “Evaluation of anomalies and neurodevelopment in children exposed to ZIKV during pregnancy”.

We appreciate your suggestions. The manuscript has been modified according to your observations, as listed below:

  1. Very short introduction section. You made a sharp introduction to the background of the study. Better start by mentioning the pandemic, the areas where the pandemic was, the way of transmission and then with the main symptoms. The prevalence in affected areas…

The introductory section has been enriched by addressing the topics suggested in lines 43 - 58 and 69 - 74.

  1. As important as the findings are, the sample is too small to invalidate previous research results. It is therefore important in a paragraph after the discussion to highlight the limitations and strengths of the study.

The wording was corrected to avoid suggesting that we are invalidating previous results in lines 257 - 260. The suggestion to highlight the strengths and limitations of the study was addressed in lines 275 - 283.

  1. The conclusions should emphasize the importance of conducting similar studies with a larger population in order to reach safe conclusions.

Thank you for your comment, the conclusions section was modified in lines 286 - 303 in response to your suggestion.

We appreciate your valuable time and attention.

Reviewer 2 Report

Dear authors, 

We appreciate undertaking a study of infants with congenital Zika syndrome born in Mexico. These infants clearly merit attention. 

The retrospective design of the study  does not allow for any conclusion on the frequency of physical anomalies or the neurodevelopmental deficits. In fact those numbers are so high that they do not allow any comparison with other series in the literature.  Also calling them minor and major anomalies seems unfortunate, neurodevelopmental deficits would be much more accurate for those that are not physical defects. 

Frankly I feel a paper with the study of the 18 children with much more detail on their deficits, as well as details on support or therapies, even a questionnaires to the mothers to reflect adaptive functioning and quality of life would be more appropriate. 

Author Response

Dear Reviewer:

We thank you for reviewing our manuscript, “Evaluation of anomalies and neurodevelopment in children exposed to ZIKV during pregnancy”.

We appreciate your suggestions, as listed below:

  1. The retrospective design of the study does not allow for any conclusion on the frequency of physical anomalies or the neurodevelopmental deficits. In fact those numbers are so high that they do not allow any comparison with other series in the literature.  Also calling them minor and major anomalies seems unfortunate, neurodevelopmental deficits would be much more accurate for those that are not physical defects. 

Your observation is very pertinent and interesting. After meeting and discussing your valuable comments, we consider that, despite being a retrospective study, the contributions are valuable because the alterations described in the infants were taken from their clinical records, and since ZIKV is a sensitive issue in our country, it is of utmost importance to make known the alterations present in infants. The CZS is new, so it is important to show all the characteristics found and to provide the basis for comparison with subsequent studies. Since the CZS is of importance and interest for different research areas such as virology, epidemiology, medicine, among others, the classification of major and minor anomalies based on the CDC definitions helps us to understand in a simpler way the impact of congenital ZIKV infection [1].

  1. Frankly I feel a paper with the study of the 18 children with much more detail on their deficits, as well as details on support or therapies, even a questionnaires to the mothers to reflect adaptive functioning and quality of life would be more appropriate. 

We agree with your observation and consider that it is of utmost importance to know the impact of congenital ZIKV infection on the life of the infant and parents, as highlighted in lines 244 - 245. Our work team has considered the study of the impact on the quality of life of the infant and the parents affected by CZS. For this reason, when this study was carried out, we worked with an expert who conducted a qualitative evaluation with the mothers of the affected children. Therefore, a publication including all these variables will be prepared in the future.

We appreciate your valuable time and attention.

Reference:

  1. Division of Birth Defects and Developmental Disabilities, N., Centers for Disease Control and Prevention Congenital Anomalies - Definitions. https://www.cdc.gov/ncbddd/birthdefects/surveillancemanual/chapters/chapter-1/chapter1-4.html (accessed March).

Round 2

Reviewer 1 Report

Dear Authors 

You fixed all the vulnerabilities in the article I pointed out to you. Congratulations!

Author Response

Dear Reviewer:

Thank you very much for your timely comments.

Reviewer 2 Report

Dear authors, 

You throw percentages that are well above what has been published and you don't compare to the other studies or comment on why your numbers are so high. You are telling the reader that more than 50% of exposed infants will develop microcephaly, spastic quadriplegia or cognitive impairment. This has not been seen before, and you should offer explanations for this, or at least comment on it. There has to be in my opinion a selection bias for those very high rates of anomalies. Or any other explanation. Where could the bias come from?  

Also, you are not telling us what determines the functional impairment. Are the children with cognitive impairment or spastic quadriplegia those with microcephaly only or mostly? How many infants with cognitive impairment did not have any structural physical or brain imaging anomalies? That would also be interesting. 

Until these surprising results and limitations are not discussed.

Author Response

Dear Reviewer,

We thank you for your review of our manuscript. The manuscript has been modified according to your, as indicated below:

  • You throw percentages that are well above what has been published and you don't compare to the other studies or comment on why your numbers are so high. You are telling the reader that more than 50% of exposed infants will develop microcephaly, spastic quadriplegia or cognitive impairment. This has not been seen before, and you should offer explanations for this or at least comment on it. There has to be in my opinion a selection bias for those very high rates of anomalies. Or any other explanation. Where could the bias come from?

We consider the observation to be entirely relevant. Our study includes only children who have a medical condition that requires them to attend an institution for children with disabilities. This limitation was indicated in the discussion section, lines 276 - 280.

  • Also, you are not telling us what determines the functional impairment. Are the children with cognitive impairment or spastic quadriplegia those with microcephaly only or mostly? How many infants with cognitive impairment did not have any structural physical or brain imaging anomalies? That would also be interesting.
  • Until these surprising results and limitations are not discussed.

Your suggestion is very interesting; however, since this is a descriptive study, our objective was to describe the findings of the children evaluated, so it is difficult for us to establish causality. We hope that in a later study we will be able to do this type of analysis that you suggest. This is indicated in lines 260 - 261.

We appreciate your valuable time and attention.